# Clustering-Based Validation Splits for Model Selection under Domain Shift

**Andrea Napoli & Paul White** *{an1g18,P.R.White}@soton.ac.uk*
*Institute of Sound and Vibration Research*
*University of Southampton, UK*

**Reviewed on OpenReview:** *https://openreview.net/forum?id=Q692COWtiD*

## Abstract

This paper considers the problem of model selection under domain shift. Motivated by principles from distributionally robust optimisation and domain adaptation theory, it is proposed that the training-validation split should maximise the distribution mismatch between the two sets. By adopting the maximum mean discrepancy (MMD) as the measure of mismatch, it is shown that the partitioning problem reduces to kernel k-means clustering. A constrained clustering algorithm, which leverages linear programming to control the size, label, and (optionally) group distributions of the splits, is presented. The algorithm does not require additional metadata, and comes with convergence guarantees. In experiments, the technique consistently outperforms alternative splitting strategies across a range of datasets and training algorithms, for both domain generalisation and unsupervised domain adaptation tasks. Analysis also shows the MMD between the training and validation sets to be well-correlated with test domain accuracy, further substantiating the validity of this approach.

## 1 Introduction

The ability for models to maintain high performance on data lying outside their training distribution, known as domain generalisation (DG), is crucial to the widespread deployment of artificial intelligence. Although extensive research has been conducted towards developing more generalisable training algorithms (Gulrajani & Lopez-Paz, 2021), significantly less focus has been given to increasing the robustness of the model selection process, despite being as integral a part of the learning problem, and indeed, just as susceptible to distribution shifts, as the fitting of the model itself.

As with model parameters, hyperparameters chosen based on in-distribution (ID) performance lack optimality guarantees on out-of-distribution (OOD) test data. Metadata-based dataset splitting, which creates OOD validation sets distinct from both the training and test data, is commonly used in this scenario, and has empirically been shown to encourage the selection of more generalisable hyperparameters (Koh et al., 2021). This paper is motivated by the principles of distributionally robust optimisation (DRO) (Rahimian & Mehrotra, 2019), which aims to optimise for worst-case performance within some uncertainty set of distributions. This suggests that the validation set should indeed be *maximally* domain shifted from the training set, while still retaining a relevant distribution. It is proposed that a worst-case dataset split which maximises domain mismatch would balance these two aims, and further encourage the selection of robust hyperparameters.

Among other considerations, the measure of mismatch should be such that the partitioning problem can be tractably solved. To this end, it is noted that performing kernel k-means clustering is equivalent to maximising the empirical maximum mean discrepancy (MMD) between clusters (weighted by cluster size). Thus, this paper proposes to perform a validation split based on kernel k-means, and presents a modified clustering algorithm for this purpose. Specifically, constraints are introduced to the cluster assignment step to control the holdout fraction (i.e., the cluster sizes), and to preserve class and (optionally) domain/group distributions; this is then formulated and solved as a linear program, providing convergence guarantees not

present in prior work. In addition, the Nyström method for low-rank approximation is employed to make the algorithm tractable for large datasets.

The proposed method provides a model selection strategy based on OOD performance that does not require additional metadata, which is not always available. It is also argued that the method is able to capture more nuances in real-world data than can be described by a single domain variable, which is another limitation of metadata-based splits. For example, in tumour identification, domain shifts can be caused by variations in sample preparation methods or patient populations. Or, in wildlife monitoring (both acoustic and visual), these can be due to differences in environmental or weather conditions, data collection equipment, or new unseen events (Koh et al., 2021). However, all this information is unlikely to be described in the available metadata, which may only state the hospital of origin or the recording location, respectively. In such cases, splitting the data along lines closer to the underlying cause of the shift, as determined by the clustering algorithm, rather than a more loosely correlated proxy, may permit a more informed model selection which can result in hyperparameters better suited to the nature of the shift.

This paper considers the two most common paradigms for learning under domain shift: domain generalisation (DG) and unsupervised domain adaptation (UDA). Both paradigms assume the data can be grouped into "domains" which share certain characteristics, and that each domain is associated with a different data distribution. Given multiple source domains, DG seeks to learn domain-invariant feature representations such that the model becomes robust to future (unknown) changes in the data. In contrast, UDA assumes unlabelled data from a specific target domain is also available during training. Although DG operates in a stricter setting, it also requires fewer assumptions about the target data – for example, UDA methods often assume minimal label shift, but this can have a significant detrimental effect on performance (Napoli & White, 2024). UDA is also less stable than DG, as it is susceptible to phenomena such as catastrophic forgetting, negative transfer, or overfitting on the target data. However, DG requires that the causal mechanisms behind the domain shift remain stable, that is, that the test domain is drawn from the same meta-distribution as the source domains. As this is difficult to ascertain in practice, test domain performance is also harder to predict. In contrast, test domain accuracy in UDA is directly coupled to domain alignment quality, given the theoretical link between the two (Redko et al., 2022).

In summary, this paper contributes the following:

- Description of a constrained clustering algorithm based on kernel k-means which can be used to perform a training-validation split in applications expected to involve domain shifts.

- Comparison of this approach with existing validation splitting strategies for a range of datasets and algorithms, in both DG and UDA settings.

- Analysis of the relationship between test domain accuracy and the MMD between the training and validation sets.

## 2 Prior work

Gulrajani & Lopez-Paz (2021) reviewed 3 criteria for model selection in a DG setting: ID accuracy (on a randomly held-out subset of each training domain); OOD accuracy on an additional domain held-out using metadata (referred to as leave-one-domain-out); and test-distribution accuracy on a held-out subset of the test domain (referred to as the oracle criterion), which can be used to provide an upper bound on performance. Where multiple validation domains are given, a DRO-style treatment of model selection which considers only the worst-performing domain is studied by Sagawa et al. (2019); Gao et al. (2023); Pfohl et al. (2022) – although this again relies on the availability of metadata.

In addition to validation accuracy, it has been suggested that a model's stability to distribution shifts should also be explicitly considered. Prior work has quantified stability in terms of the expected calibration error (the average deviation between accuracy and confidence) (Wald et al., 2021); the MMD between features from different domains (Lyu et al., 2023); or the average variation of each feature between domains (Ye et al., 2021).

It has also been proposed to induce a domain shift between the training and validation sets by performing mix-up augmentation on the held-out data (Lu et al., 2023). This links to a range of general robustness benchmarks, where the evaluation sets have been subjected to various synthetic transformations. For example, visual corruptions and perturbations (Hendrycks & Dietterich, 2019), stylisation (Geirhos et al., 2018), the addition of spurious cues (Li et al., 2022), and adversarial filtration (Hendrycks et al., 2019) have all been employed; a more complete review of this approach is given in Koh et al. (2021).

Non-random data splits have previously been used to produce domain-shifted evaluation sets, but no prior work has investigated these in a model selection context. Søgaard et al. (2021) proposed an adversarial splitting heuristic based on k-nearest neighbours and the Wasserstein distance between term count distributions of text documents, but this is not generally applicable to features in $\mathbb{R}^n$, and no attempt was made to control the label distribution. Adversarial data splits have also been used in meta-learning to improve DG (Gu et al., 2023; Wang et al., 2024). Recently, Napoli & White (2025) proposed a data selection algorithm which can be used to choose a validation set closer to the test domain in the UDA setting. However, this is not applicable if the split needs to be decided before the test domain is accessible, e.g., for test-time adaptation.

Finally, with an approach most similar to ours, Wecker et al. (2020) proposed an evaluation split based on k-means clustering, with constraints to control for cluster size and label distribution. Our paper develops on this work in a number of ways. Firstly, the motivation of Wecker et al. (2020) is merely to create more challenging data splits, and is not applied to improving model selection nor theoretically connected to distributional robustness. Secondly, Wecker et al. (2020) impose the constraints using a greedy strategy, which are susceptible to local minima. In contrast, we solve a constrained assignment problem using linear programming, which avoids this effect and results in better quality solutions. We also generalise the label-wise constraints to domains for multi-domain data. Finally, our theoretical analysis motivates a kernelised version of this algorithm, thereby exploiting the relation between the kernelised objective (which we link to the MMD) and validation accuracy.

The theoretical relation between the MMD and kernel k-means clustering has also been derived in other contexts. For example, França et al. (2017) obtain similar results from the viewpoint of generalised energy statistics, while Ohl et al. (2024) use the relation to improve training of unsupervised decision trees.

## 3 Preliminaries and notation

Let $S = \{(x_1, y_1, d_1), \ldots, (x_n, y_n, d_n)\}$ be the development set consisting of input-label-domain triplets over $\mathcal{X} \times \mathcal{Y} \times \mathcal{D}^S$. Similarly, let $E$ be the evaluation set over $\mathcal{X} \times \mathcal{Y} \times \mathcal{D}^E$, where $\mathcal{D}^S \cap \mathcal{D}^E = \emptyset$ (i.e., the development and evaluation domains are disjoint), and let $\mathcal{Y}$, $\mathcal{D}^S$ and $\mathcal{D}^E$ be finite sets. For ease of notation, subscripts are used on sets to simplify set-builder notation, in two ways. Firstly, "slices" of a set are denoted using capitals, for example:

$$S_X = \{x : (x, y, d) \in S\}$$

$$S_{YD} = \{(y, d) : (x, y, d) \in S\}.$$

Additionally, a predicate can be specified to restrict the set to samples satisfying a condition. For example, to denote only inputs associated with a specific class $y'$:

$$S_{X,Y=y'} = \{x : (x, y, d) \in S \land y = y'\}.$$

In short, the goal of DG is to use $S$ to produce a model $\theta : \mathcal{X} \to \mathcal{Y}$ that performs well on $E$. $\theta$ comprises a featuriser $\theta_F : \mathcal{X} \to \mathcal{F}$ and label classifier $\theta_C : \mathcal{F} \to \mathcal{Y}$, such that $\theta = \theta_C \circ \theta_F$. In order to tune hyperparameters, $S$ must be partitioned into training and validation sets, $T$ and $V$ respectively. A number of models $\Theta = \{\theta_1, \ldots, \theta_m\}$ are trained on $T$ using different hyperparameters; the "best" model is then selected, according to

$$\theta^* = \arg\min_{\theta \in \Theta} R(\theta, V), \tag{1}$$

where $R(\theta, V)$ is the error of $\theta$ on $V$, and $\theta^*$ is evaluated on $E$.

### 3.1 Theoretical motivation

DRO (Rahimian & Mehrotra, 2019; Sagawa et al., 2019) is a well-established and theoretically grounded paradigm for obtaining more generalisable solutions to optimisation problems. Instead of reducing overall error, DRO adopts a minimax strategy which minimises the worst-case error over some uncertainty set $\mathcal{Q}$. By defining $\mathcal{Q}$ as the possible (suitable) partitions of $S$, the model selection problem becomes

$$\min_{\theta \in \Theta} \max_{\{T,V\} \in \mathcal{Q}} R(\theta, V). \tag{2}$$

This could be solved directly via an expensive alternating optimisation process (Gu et al., 2023). However, we propose instead to find $\arg\max_{T,V} R(\theta, V)$, the worst-case partition in the sense of $R$, in a single step by solving a surrogate problem based on the well-known relationship between $R(\theta, V)$ and the mismatch between $T$ and $V$, which has been fundamental to domain adaptation theory (Ben-David et al., 2006; 2010). By using the MMD to measure this mismatch, the partitioning problem can be efficiently solved through clustering, as will be shown in Section 4. The theoretical relation between $\mathrm{MMD}(T, V)$ and $R(\theta, V)$ is given by the generalisation error bound (Redko et al., 2022, Theorem 36)

$$R(\theta, V) \le R(\theta, T) + \mathrm{MMD}(T, V) + \lambda \tag{3}$$

where $\lambda$ is a term depending on the capacity of the hypothesis space and the combined performance of an ideal model on both $T$ and $V$. A linear correlation has also been observed empirically (Napoli & White, 2025). Although maximising the MMD is not guaranteed to maximise this bound due to the dependence of $R(\theta, T)$ on the partitioning, the empirical analysis in Section 5.5 shows that this term remains fairly stable in practice.

Equation (3) is quite general and not strictly limited to covariate shift. However, the robustness of the bound relies on the ability of the model to simultaneously perform well on both $T$ and $V$ (through $\lambda$), and this can be affected by the presence of conditional shifts. There are also assumptions made about both the (joint) data distributions and the hypothesis class of models – we refer to Redko et al. (2022) for formal statements of these – and these ultimately define the reproducing kernel Hilbert space in which the MMD in the bound is measured. Thus, by symmetry, the choice of kernel shapes the underlying assumptions about the model and data, and ultimately determines whether the MMD being optimised is predictive of the discrepancy between the training and validation losses.

## 4 Method

This section defines the constraints needed to ensure an appropriate split, formulates the partitioning problem, and describes an algorithm to solve the optimisation.

$T$ and $V$ should be of sizes determined by a user-defined holdout fraction $h$ satisfying $0 < h < 1$, and have equivalent class distributions. This can be achieved by constraining the size of each label group in $V$ to $h$ times the size of the corresponding group in $S$:

$$|V_{Y=g}| = h\,|S_{Y=g}|, \quad \forall g \in \mathcal{Y}. \tag{4}$$

This constraint ensures that there are sufficient examples from each class in both $T$ and $V$ to properly train and validate the models. Without it, the data may have a tendency to cluster by class, which may result in certain classes being placed entirely in one set.

It may also be necessary or desirable to control domain distributions. For example, certain training algorithms may require that the domains in $T$ be uniformly represented to avoid overfitting; controlling the distributions of validation groups has also been suggested to reduce noise in the hyperparameter tuning process (Sagawa et al., 2019). Finally, in the multi-task learning setting (i.e., some or all of the domains correspond to different tasks, with separate validation metrics), the domain split-ratio should also be controlled: this both prevents bias towards a specific task, and also ensures that each task has sufficient representation in the validation set to reliably estimate the corresponding validation metric. In these cases, the constraints

should be taken over all $(y, d)$ pairs instead:

$$\left| V_{(Y,D)=g} \right| = h \left| S_{(Y,D)=g} \right|, \quad \forall g \in \mathcal{Y} \times \mathcal{D}^S. \tag{5}$$

For the remainder of this section, the latter set of constraints (5) are assumed, although (4) can easily be substituted if desired (as relaxing the constraints will increase the clustering objective), or if domain labels are unavailable.

The objective is to maximise the discrepancy between $T$ and $V$, measured using the MMD. Given the potentially high dimensionality of $\mathcal{X}$, doing this directly in the input space can be impractical. Therefore, it is proposed to instead measure the MMD between the distributions of feature sets $\theta_F[T_X]$ and $\theta_F[V_X]$, where $\theta_F[\cdot]$ denotes the image under $\theta_F$. Moreover, these intermediate features are those most highly correlated with the model's outputs, meaning this is the representation space for which the MMD is most predictive of classification error. For further intuition, consider that a large MMD in this space implies a domain shift that the model is not invariant to (and thus is more likely to cause problems) – hence why this is also the representation space normally targeted by feature alignment methods (Gulrajani & Lopez-Paz, 2021). For models where no intermediate representations are available, the input representations can be used directly, or an alternative feature extractor or dimensionality reduction technique can be used. Prior knowledge on the expected nature of the domain shift could also be incorporated into the clustering features; a detailed discussion this is given in Appendix A.

Assume $\theta_F[T_X]$ and $\theta_F[V_X]$ are samples from distributions $\mathbb{P}_T$, $\mathbb{P}_V$ over $\mathcal{F}$. A positive-definite kernel $\kappa : \mathcal{F} \times \mathcal{F} \to \mathbb{R}$ induces a unique reproducing kernel Hilbert space (RKHS) $\mathcal{H}$ on $\mathcal{F}$, along with a mapping $\phi : \mathcal{F} \to \mathcal{H}$. The empirical mean map of $\mathbb{P}_T$ (and analogously for $\mathbb{P}_V$) in $\mathcal{H}$ is given by

$$\mu_{\mathbb{P}_T} = \frac{1}{|T_X|} \sum_{f \in \theta_F[T_X]} \phi(f). \tag{6}$$

The MMD can then be estimated as the distance between means of samples embedded in $\mathcal{H}$:

$$\mathrm{MMD}\left(\mathbb{P}_T, \mathbb{P}_V\right) = \left\| \mu_{\mathbb{P}_T} - \mu_{\mathbb{P}_V} \right\|_{\mathcal{H}}. \tag{7}$$

The partitioning problem can now be formulated as

$$\arg \max_{T,V} \left\| \mu_{\mathbb{P}_T} - \mu_{\mathbb{P}_V} \right\|_{\mathcal{H}} \quad \text{subject to (5).} \tag{8}$$

This is equivalent to performing kernel k-means clustering with $k = 2$ (i.e., using one cluster for each of $T$ and $V$), subject to the same constraints:

$$\arg \min_{T,V} \Psi(T, V) \quad \text{subject to (5),} \tag{9}$$

where $\Psi(T, V) = \mathrm{SSq}(T_X) + \mathrm{SSq}(V_X)$ is the standard kernel k-means objective function and

$$\mathrm{SSq}(T_X) = \sum_{f \in \theta_F[T_X]} \| \phi(f) - \mu_{\mathbb{P}_T} \|_{\mathcal{H}}^2 \tag{10}$$

is the sum of squared deviations of a set of points from their centroid (in feature space).

**Theorem 1.** *Problems (8) and (9) are equivalent.*

*Proof sketch.* The equivalence can be derived by applying an ANOVA sum-of-squares decomposition, followed by a substitution based on the polarisation identity. The resulting identity is

$$\| \mu_{\mathbb{P}_T} - \mu_{\mathbb{P}_V} \|_{\mathcal{H}}^2 = \frac{|S_X|}{|T_X||V_X|} (\mathrm{SSq}(S_X) - \Psi(T, V)). \tag{11}$$

For a full derivation, see Appendix B. As $S_X$ is constant with respect to the cluster allocations, and the cluster sizes are fixed by the constraint, these terms can all be dropped from the objective function. Hence, (8) and (9) are equivalent. □

Problem (9) can be solved by applying a variation of Lloyd's algorithm (Chitta et al., 2014).

**Algorithm 1** Constrained kernel k-means clustering

Given an initial set of assignments, alternate between 2 steps until convergence or max iterations reached:

1. **Distance Update** (maximisation). Compute the distance matrix $D \in \mathbb{R}^{n \times 2}$ from each point to the centroid of each cluster using the kernel trick. For large datasets, use the Nyström method (Chitta et al., 2014) to reduce the complexity of the kernel computations from $O(n^2)$ to $O(qn)$, by computing only a randomly selected submatrix of the full kernel, of size $q \times n$.

2. **Constrained Assignment** (expectation). Compute the one-hot cluster assignment matrix $U \in \{0,1\}^{n \times 2}$ that assigns each point to exactly one of the two clusters. $U$ is the solution to the binary linear program (LP):

$$\arg\min_U \sum_{i,j} U_{ij} D_{ij} \tag{12}$$

subject to

$$U_{ij} \in \{0,1\} \quad \text{for } i \in \{1,\dots,n\}; \ j \in \{1,2\} \tag{13}$$

$$\sum_j U_{ij} = 1 \quad \text{for } i \in \{1,\dots,n\} \tag{14}$$

$$\sum_{i|(Y_i,D_i)=g} U_{ij} = \text{round}\left(h\left|S_{(Y,D)=g}\right|\right) \quad \text{for } g \in \mathcal{Y} \times \mathcal{D}^S; \ j = 1 \text{ or } 2. \tag{15}$$

(14) ensures that each point is assigned to only one cluster. The disjunctive constraint (15) enforces (5), and indicates that $j$ can take a value of *either* 1 or 2. The disjunction arises as (5) is independent of the cluster indices, i.e., it does not matter which index is designated as the validation set. Which option has lower cost depends on the initialisation of the centroids. As there are only 2 clusters, the easiest way to approach this is simply to solve 2 LPs, one for each value of $j$, and then select the lower-cost solution.

The constraints satisfy Hoffman's sufficient conditions for total unimodularity (Heller & Tompkins, 1956) (in particular, it can be seen that (14) and (15) form two disjoint sets of constraints, and every element of $U$ is referenced at most once in each set). The consequence is that the LP will always have integer solutions, without having to enforce them explicitly. This means the binary constraint (13) can be relaxed to

$$0 \le U_{ij} \le 1 \text{ for } i \in \{1,\dots,n\}; \ j \in \{1,2\} \tag{16}$$

and the problem can be solved without integer constraints. To enforce soft constraints, (15) can be replaced by the inequality

$$\text{round}\left(h\left(1-\tau_g\right)\left|S_{(Y,D)=g}\right|\right) \ \le \ \sum_{i|(Y_i,D_i)=g} U_{ij} \le \text{round}\left(h\left(1+\tau_g\right)\left|S_{(Y,D)=g}\right|\right) \text{ for } g \in \mathcal{Y} \times \mathcal{D}^S; \ j = 1 \text{ or } 2, \tag{17}$$

where $\tau_g$ is the relative tolerance for constraint associated to $g$.

**Proposition 2.** *Algorithm 1 converges to a locally optimal partitioning in a finite number of iterations.*

*Proof sketch.* Note that $\Psi(T,V)$ is bounded below by 0 and is also non-increasing, since both the centroid updates and cluster assignments are (or can be interpreted as) optimisation problems which share the same objective function as (9), can be solved globally at each iteration, and do not violate any of the constraints. Note also that only a finite number of partitionings are possible, meaning $\Psi(T,V)$ can only decrease a finite number of times. Therefore, convergence in finite time is guaranteed. For a full proof, see Bradley et al. (2000). □

Although Proposition 2 provides a trivial upper bound for the number of iterations required until convergence, Lloyd's algorithm is known to be fast in practice, and the runtime is often considered to be linear in the

number of datapoints based on empirical observations (Cordeiro de Amorim & Makarenkov, 2023). For further theoretical analysis on the number of iterations required, see, e.g., Arthur et al. (2009); Arthur & Vassilvitskii (2006).

### 4.1  Kernel choice

As noted in Section 3.1, the choice of kernel implies certain assumptions about the data. In some applications (e.g., physical modelling), known feature constraints naturally guide the kernel choice. In our case, however, only relatively weak geometric assumptions can be made. The kernel should also be *characteristic* to ensure the MMD can distinguish between any two distributions (Sriperumbudur et al., 2009). However, we note that, unlike kernel methods which make use of the kernel distances directly, kernel k-means clustering only *compares between* kernel distances when assigning datapoints, which we would argue makes the solution less sensitive to kernel choice or kernel hyperparameters. Additionally, the cluster size constraints restricts the solution space, which further desensitises the solution to kernel choice.

For the experiments in the subsequent section, we opt to use a Gaussian radial basis function (RBF) kernel $\kappa(x, y) = e^{-\gamma \|x-y\|^2}$, which is a convenient and popular choice for MMD estimation (Sriperumbudur et al., 2009). Perhaps more relevantly, we also note that the correlation between the MMD (with an RBF kernel) and $R(\theta, V)$ has been supported by empirical observations (Napoli & White, 2025).

## 5  Experiments

The benefits of any new model selection method can only be verified when the oracle criterion suggests there is "room for improvement" over a basic random split, i.e., there is a performance gap between the two. Thus, the experiments described in this section are set up to reflect this scenario (the limitations of this are discussed further in Section 6). For example, it is noticed that UDA tends to exhibit a larger gap than DG, and this is especially pronounced (perhaps unsurprisingly) for adversarial algorithms, which tend to be more sensitive to hyperparameter choices.

Two batches of experiments are run, to reflect both the UDA and DG settings. Each batch comprises an identical training setup applied to 3 different datasets. For the DG experiments, models are trained using the CORAL algorithm (Sun & Saenko, 2016), with the clustering performed using constraints (4). For the UDA experiments, the DANN algorithm (Ganin et al., 2015) is used to adapt to an additional, unlabelled subset of test domain samples, as well as to align the training domains to each other. As DANN was observed to be more sensitive to domain imbalances, the validation split is set to preserve domain distributions, i.e., using constraints (5).

In this work, we fix the RBF bandwidth to $\gamma = 1$. To assess the sensitivity of our results to the choice of kernel, we also test a linear kernel $\kappa(x, y) = x^T y$, which is equivalent to linear k-means clustering.

All feature extractors are finetuned on the entirety of $S$ before the features are computed for the clustering, regardless of pretraining. Experiments are conducted using the DomainBed framework (Gulrajani & Lopez-Paz, 2021). This means all-but-one of the domains are placed in the development set, and the remaining "evaluation" domain is randomly split into a UDA set (for adaptation, unused for the DG experiments), and an independent test set used to determine final accuracy values. Every domain is tested 3 times for reproducibility, each time with a different random seed for model initialisation, hyperparameter search and other stochastic variables. The reported accuracy values are averages over all domains and repeats. Further training and hyperparameter search details are given in Appendix C; unless otherwise stated, the remaining details all follow the Domainbed default options. The Gurobi Optimizer (Gurobi Optimization LLC, 2023) is used to solve the LPs. In total, the experiments involve training 5,160 models, requiring around 100 GPU-days of computation.

### 5.1  Datasets

The datasets represent a range of domain shift problems encompassing both image and audio classification tasks. In addition to the covariate shifts which occur across all datasets, the two ecological datasets (Hump-

backs and TerraIncognita) also contain significant conditional shifts due to the open-set and sometimes annotator-dependent nature of wildlife monitoring data. With the exception of Camelyon17, the datasets are all small enough that the entire kernel matrix can be computed. So, Camelyon17 is the only dataset for which the Nyström method is applied.

**Camelyon17-WILDS** (Bándi et al., 2019; Koh et al., 2021) tumour detection in tissue samples across 5 hospitals, 2 classes and 455,954 samples. In keeping with the WILDS setup, the model in this case is trained from scratch rather than using pretrained weights. However, note that these results still cannot be compared directly with results from WILDS, as the DomainBed setup does not match exactly. License: CC0.

**Humpbacks** (Napoli & White, 2023) detection of humpback whale vocalisations across 4 recording locations, 2 classes and 43,385 samples. This is the only dataset not to use the ResNet-18 architecture, and instead uses a custom CNN architecture and acoustic front-end described in Appendix C. License: Proprietary.

**SVIRO** (Dias Da Cruz et al., 2020) classification of vehicle rear seat occupancy across 10 car models, 7 classes and a balanced subset of 24,500 samples of the original dataset. License: CC BY-NC-SA 4.0.

**VLCS** (Fang et al., 2013) object classification across 4 image datasets, 5 classes and 10,729 samples. License: unknown.

**PACS** (Li et al., 2017) object classification across 4 image styles (photos, art, cartoons, and sketches), 7 classes and 9,991 samples. License: unknown.

**Terra Incognita** (Beery et al., 2018) classification of wild animals across 4 camera trap locations, 10 classes and 24,788 samples. License: CDLA-Permissive 1.0.

## 5.2 Results

In total, 6 model selection methods are compared. These are: the random split; the leave-one-domain-out split; the test domain (oracle) validation set; a random split followed by mix-up augmentation on the validation set (Lu et al., 2023); and cluster-based splits using linear and kernel k-means clustering. The random and random-plus-mix-up splits are stratified by domain. The validation and test accuracies are class-balanced for all methods.

The results are shown in Table 1, along with standard errors (the standard errors are as computed by DomainBed, and capture variability in the overall experimental run, including random seeds and across domains). In the following sections, a 95% confidence level is used when verifying whether two values have a statistically significant difference, which corresponds to non-overlapping confidence intervals of 1.96 times the standard error (assuming normally distributed errors).

Table 1: Average test domain accuracies for all datasets and model selection criteria.

| Split type | DG experiments | | | UDA experiments | | | Raw average | Normalised average |
|---|---|---|---|---|---|---|---|---|
| | Camelyon | Humpbacks | SVIRO | VLCS | PACS | TerraInc | | |
| Random | 84.0 ± 1.0 | 76.4 ± 2.1 | 98.1 ± 0.2 | 70.7 ± 2.9 | 80.3 ± 0.3 | 38.7 ± 2.9 | 74.7 ± 0.8 | 0.0 ± 11.8 |
| Leave-one-out | 85.6 ± 1.0 | 77.3 ± 1.7 | 98.6 ± 0.0 | 71.3 ± 3.5 | 83.7 ± 0.6 | 37.3 ± 2.5 | 75.6 ± 0.8 | 28.2 ± 11.7 |
| Linear k-means | 87.2 ± 0.2 | 78.0 ± 1.8 | 98.4 ± 0.0 | 76.9 ± 0.1 | 82.7 ± 0.2 | 43.0 ± 2.0 | 77.7 ± 0.5 | 55.2 ± 5.9 |
| RBF k-means | 87.3 ± 0.7 | 78.3 ± 0.6 | 98.6 ± 0.2 | 75.2 ± 0.8 | 82.3 ± 0.8 | 40.1 ± 2.1 | 77.0 ± 0.4 | 46.9 ± 7.6 |
| Mix-up | 85.1 ± 0.4 | 76.1 ± 1.2 | 98.2 ± 0.1 | 73.8 ± 1.7 | 80.5 ± 0.6 | 37.3 ± 2.9 | 75.2 ± 0.6 | 10.4 ± 8.9 |
| Oracle | 88.3 ± 0.6 | 85.4 ± 1.5 | 99.1 ± 0.1 | 77.6 ± 0.5 | 84.4 ± 0.4 | 45.8 ± 1.0 | 80.1 ± 0.3 | 100.0 ± 5.1 |

The range of possible performance improvement differs by dataset, as determined by the gap between the random split and oracle criterion; accuracy values should be considered relative to this scale when averaging across datasets. Therefore, a column of average normalised values is also shown, where each dataset is shifted and scaled such that the random split has a value of 0 and the oracle a value of 100.

Where the validation set comprises multiple domains, model selection is based on average validation accuracy across these domains, as is the DomainBed default. Results based on worst-domain accuracy are also shown in Appendix D. In theory, worst-domain accuracy should provide an additional layer of distributional robustness

with respect to the domain labels (Sagawa et al., 2019). However, no significant difference in test accuracy overall is observed if worst-case validation accuracy is used, although the standard error does increase. It is possible that the uncertainty set $\mathcal{Q} = \mathcal{D}^S$ has too few degrees of freedom for DRO to confer any meaningful robustness in this case.

On average, the cluster-based splits provide a net absolute accuracy gain of around 3 percentage points compared to the random split, and 2 percentage points gain compared to leave-one-domain out validation. In relative terms, clustering is observed to close around 50% of the gap between the random split and oracle criterion, compared to 28% for leave-one-domain-out validation and 10% for mix-up. Overall, performance is slightly higher using the linear kernel than with the RBF kernel, although this is within margin of error. It is possible that the latter may be improved with more careful tuning of $\gamma$. It is noted that both clustering methods are also within margin of error of leave-one-domain-out validation.

### 5.3 Ablation study on VLCS

An ablation study conducted on the VLCS dataset is shown in Table 2. This shows the effects of finetuning the feature extractor before clustering, the Nyström approximation, and the different constraint sets (4) and (5). For this dataset, use of the Nyström approximation, as well as additional finetuning of $\theta_F$, are not observed to have significant effects on test accuracy. For the linear kernel, clustering with constraints (4) performs significantly lower than using constraints (5), however, for the RBF kernel, this difference is not observed.

Table 2: Ablation study on VLCS.

| Kernel/ Split type | Finetuning $\theta_F$ | Constraint groups | Full/ Nyström kernel | Accuracy (%) |
|---|---|---|---|---|
| Linear | True | $g \in \mathcal{Y} \times \mathcal{D}$ | Nyström | $76.4 \pm 0.4$ |
| RBF | True | $g \in \mathcal{Y} \times \mathcal{D}$ | Nyström | $76.2 \pm 0.6$ |
| Linear | True | $g \in \mathcal{Y}$ | Full | $72.4 \pm 1.9$ |
| RBF | True | $g \in \mathcal{Y}$ | Full | $75.1 \pm 0.9$ |
| Linear | True | $g \in \mathcal{Y} \times \mathcal{D}$ | Full | $76.9 \pm 0.1$ |
| RBF | True | $g \in \mathcal{Y} \times \mathcal{D}$ | Full | $75.2 \pm 0.8$ |
| Linear | False | $g \in \mathcal{Y} \times \mathcal{D}$ | Full | $75.6 \pm 0.8$ |
| RBF | False | $g \in \mathcal{Y} \times \mathcal{D}$ | Full | $76.4 \pm 0.5$ |
| Mix-up | True | N/A | N/A | $73.8 \pm 1.7$ |
| Leave-one-domain-out | True | N/A | N/A | $71.3 \pm 3.5$ |
| Random | True | N/A | N/A | $70.7 \pm 2.9$ |
| Oracle | True | N/A | N/A | $77.6 \pm 0.5$ |

### 5.4 MMD analysis

As stated in Section 1, the motivation for cluster-based splits is the notion that increasing the MMD between the training and validation sets increases test domain accuracy. To provide empirical support for this, these two variables are plotted against each other in Figure 1, for each dataset. As for the clustering, an RBF kernel is used with $\gamma = 1$. Again, the Nyström method is used to estimate the MMD for the Camelyon17 dataset due to its size. Each subplot in Figure 1 shows a different dataset, and each point in a subplot represents one of the 5 split types (not including the oracle), averaged across the domains and 3 repeats. Standard errors were sufficiently small (i.e., at least 2 order of magnitude smaller than the data range for all subplots) that error bars were not included. The correlation coefficients and associated $p$-values are also shown. As only monotonic associations are being tested for, Spearman's rank correlation $\rho$ is used.

The leave-one-domain-out method often produced large outlier values for the MMD, which can be seen in Figure 1 for all datasets other than Camelyon17 and Humpbacks. The reason for these outliers is unclear. Nonetheless, the correlation for all datasets, with the exception of PACS, is positive.

The correlation is clearest for Camelyon17, possibly because the dataset is large enough that a low-noise estimate of the MMD is possible. The results for this dataset are given in tabular form in Table 3, along with an additional ablation study comparing the different clustering constraints (4) and (5). It can be seen that

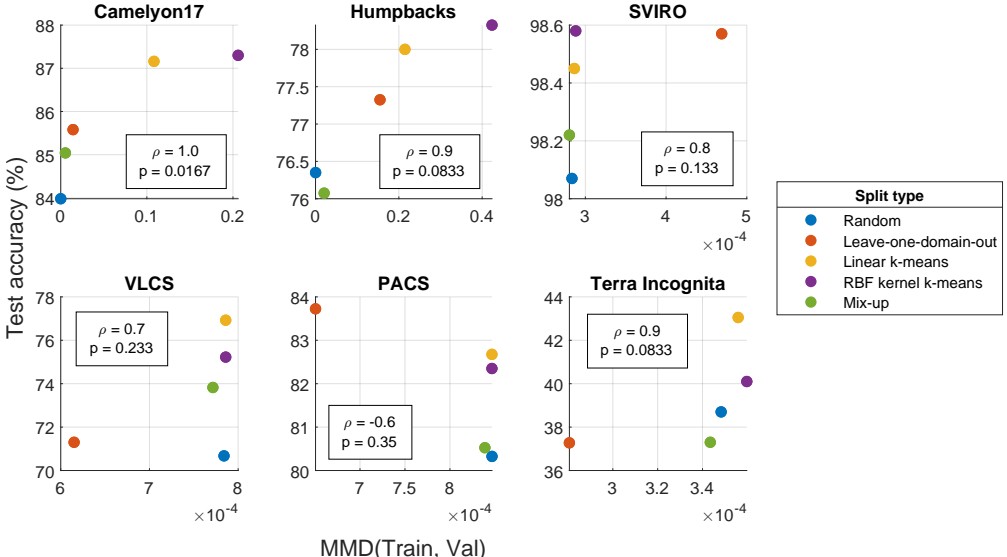

Figure 1: The MMD between the training and validation sets versus test domain accuracy, by dataset.

Table 3: The MMD between the training and validation sets versus test domain accuracy by split type for Camelyon17.

| Split type (Constraint) | Accuracy (%) | MMD ×1000 |
|---|---|---|
| Random | 84.0 ± 1.0 | 0.01 ± 0.0003 |
| Mix-up | 85.1 ± 0.4 | 5.4 ± 0.2 |
| Leave-one-domain-out | 85.6 ± 1.0 | 14.3 ± 0.8 |
| Linear k-means (5) | 86.5 ± 1.3 | 97.0 ± 6.2 |
| RBF kernel k-means (5) | 85.6 ± 1.3 | 183.1 ± 2.0 |
| Linear k-means (4) | 87.2 ± 0.2 | 108.3 ± 2.6 |
| RBF kernel k-means (4) | 87.3 ± 0.7 | 206.0 ± 3.4 |

the additional constraint on the clustering (i.e., taking $g \in \mathcal{Y} \times \mathcal{D}$) reduces the optimised objective function value (as would be expected), and this also corresponds to a reduction in test domain model accuracy.

The evidence in this section supports the proposition that the validation split should be attempting to maximise the MMD between the training and validation sets, and that this is more effective than using the same metadata-based splitting rule as the test split, as the leave-one-domain-out split intends to do.

### 5.5 Assessing the effect of the changing training distribution

As evidenced in (3), the use of non-random data splitting introduces a confounding variable to the experiments: as both the training and validation distributions are dependent on the split, it is possible the test accuracy is being influenced by the model training, as well as the validation. To support the claim that cluster-based splitting results in more generalisable model selection, it is necessary to decouple the effects of these interventions, and show that it is indeed the improved model validation, and not the training, that is improving test accuracy.

As non-random data splitting inherently changes the training distribution, the effect of the model selection cannot be isolated from that of the training. However, the inverse – that is, varying the training distribution by changing the split type, but keeping the validation set constant – can be achieved, if the oracle selection criterion is used.

If the test accuracy remains constant across split types, this would be sufficient to verify the hypothesis that the performance improvements are due to the model selection, without having to make any additional

assumptions. If the accuracy is lower than the random split, the hypothesis can still be confirmed (which then implies the model selection is additionally compensating for this reduction), as long as the effects of training and validation robustness on test accuracy are assumed to be additive (i.e., any interaction effects are minor). If anything, the non-random splits *would* be expected to underperform, since coverage of the overall data distribution by the training set is being reduced – and this would be expected to be detrimental to generalisation power. The results are given in Table 4.

Table 4: Test accuracy using the oracle criterion, for training sets induced by the different split types.

| Split type | DG experiments | | | UDA experiments | | | Average |
|---|---|---|---|---|---|---|---|
| | Camelyon17 | Humpbacks | SVIRO | VLCS | PACS | TerraInc | |
| Random | $88.3 \pm 0.6$ | $85.4 \pm 1.5$ | $99.1 \pm 0.1$ | $77.6 \pm 0.5$ | $84.4 \pm 0.4$ | $45.8 \pm 1.0$ | $80.1 \pm 0.3$ |
| Leave-one-domain-out | $85.4 \pm 0.4$ | $82.1 \pm 1.3$ | $99.0 \pm 0.1$ | $73.8 \pm 0.3$ | $82.2 \pm 0.5$ | $38.9 \pm 1.2$ | $76.9 \pm 0.3$ |
| Linear k-means | $88.0 \pm 0.3$ | $85.4 \pm 0.8$ | $99.1 \pm 0.1$ | $77.0 \pm 0.3$ | $85.7 \pm 0.3$ | $47.1 \pm 0.9$ | $80.4 \pm 0.2$ |
| RBF kernel k-means | $88.6 \pm 0.6$ | $85.0 \pm 0.7$ | $99.2 \pm 0.1$ | $77.9 \pm 0.5$ | $85.7 \pm 0.3$ | $44.5 \pm 0.6$ | $80.2 \pm 0.2$ |
| Mix-up | $88.3 \pm 0.6$ | $85.6 \pm 1.6$ | $99.2 \pm 0.1$ | $77.3 \pm 0.5$ | $84.4 \pm 0.5$ | $44.6 \pm 0.9$ | $79.9 \pm 0.3$ |

These results show that performing a cluster-based split has no significant effect on the generalisation power of a model when the hyperparameters are being chosen via the oracle criterion. Therefore, it can be concluded that the performance improvements of cluster-based splitting seen in Table 1 come entirely from the model selection, and *not* from the training. On the other hand, test accuracy for the leave-one-domain-out split does significantly reduce. This finding may help to explain why metadata-based splits have been found to underperform random splits in some works (Gulrajani & Lopez-Paz, 2021): the increased robustness of OOD validation is not enough to counterbalance the reduced robustness of training on fewer domains.

## 5.6 Convergence behaviour of Algorithm 1

As mentioned previously, the proposed split method is similar in principle to the one of Wecker et al. (2020). One important implementational distinction is the manner by which the constraints are enforced. We have claimed that the greedy approach of Wecker et al. (2020) is susceptible to local minima and thus tends to converge to inferior solutions. To support this claim, this section presents an experiment to compare the quality of the optima attained by the two methods. We use the official implementation of Wecker et al. (2020), which uses linear k-means with label-only constraints (4), and configure our method similarly for fair comparison. For each domain in the VLCS dataset, we run both algorithms 20 times with different random initialisations, and $h = 0.5$. The attained clustering objectives $\Psi(T, V)$ are shown in Figure 2. It can be seen that our method has significantly less spread than the one of Wecker et al. (2020) (around 1000 times less for Domain 3, 100 times less for Domains 1 and 4, and 1.8 times less for Domain 2), and the mean losses also tend to be lower. These findings suggest that the greedy method is indeed getting trapped in local minima.

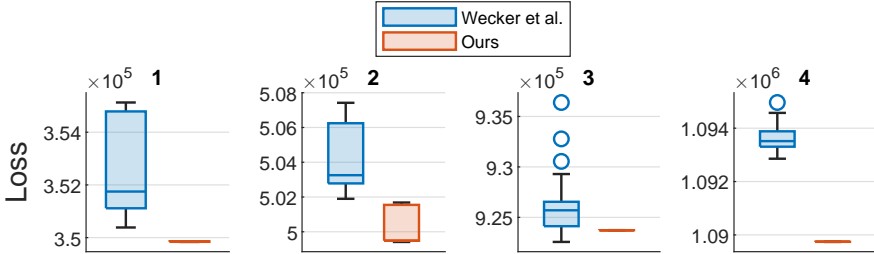

Figure 2: The attained loss of our clustering algorithm vs that of Wecker et al. (2020), on the Humpbacks dataset, by domain.

## 5.7 Assessing the effect of the number of classes

As can be inferred from the problem formulation, a large number of classes may limit the effectiveness of the method. Specifically, with more classes, the partitioning problem is less flexible due to the increased

number of clustering constraints, which reduces the upper bound on the MMD which can be achieved. For example, in an extreme case with only two examples per class, a random split (stratified by class) and a clustering-based split would be equivalent. Thus, this section presents an experiment to analyse the effect of the number of classes on $\text{MMD}(T, V)$. Specifically, using a Humpbacks subset of 8,000 examples, we generate synthetic class labels using another round of kernel k-means clustering, with a constraint to ensure equal class sizes. Then, we perform a kernel k-means data split using the synthetic labels, and the label-only constraints (4). Table 5 shows how the MMDs change as the number of classes varies between 1 and 500. We also report the MMDs for two of the baselines: the random split and the metadata-based split.

Table 5: The MMDs of the kernel k-means clustering data split by number of classes, and two baselines.

| Number of classes/ Baselines | $\text{MMD}(T, V)$ | $\text{MMD}(T, E)$ | $\text{MMD}(V, E)$ |
|---|---|---|---|
| 1 | $0.221 \pm 0.016$ | $0.152 \pm 0.042$ | $0.148 \pm 0.040$ |
| 2 | $0.187 \pm 0.007$ | $0.154 \pm 0.048$ | $0.129 \pm 0.048$ |
| 5 | $0.177 \pm 0.007$ | $0.121 \pm 0.037$ | $0.157 \pm 0.034$ |
| 10 | $0.161 \pm 0.006$ | $0.132 \pm 0.020$ | $0.138 \pm 0.021$ |
| 20 | $0.159 \pm 0.005$ | $0.132 \pm 0.021$ | $0.137 \pm 0.019$ |
| 50 | $0.145 \pm 0.002$ | $0.137 \pm 0.031$ | $0.125 \pm 0.031$ |
| 100 | $0.138 \pm 0.004$ | $0.119 \pm 0.019$ | $0.140 \pm 0.017$ |
| 500 | $0.129 \pm 0.001$ | $0.123 \pm 0.027$ | $0.130 \pm 0.027$ |
| Random | $0.001 \pm 0.000$ | $0.095 \pm 0.064$ | $0.095 \pm 0.064$ |
| Leave-one-domain-out | $0.052 \pm 0.034$ | $0.053 \pm 0.036$ | $0.135 \pm 0.087$ |

To determine a threshold for the maximum number of classes for which a clustering-based split is still valid, we can consider the range for which $\text{MMD}(T, V)$ is higher than $\text{MMD}(T, E)$ and $\text{MMD}(V, E)$, since this is the assumption made by DRO. We consider both $\text{MMD}(T, E)$ and $\text{MMD}(V, E)$ since the data split is symmetric (it does not matter which way around $T$ and $V$ are labelled). Note that $\text{MMD}(T, E)$ and $\text{MMD}(V, E)$ also tend to reduce with $\text{MMD}(T, V)$ due to the triangle inequality. In this case, it can be seen that this threshold occurs at around 100 classes. Although, it is noted that even for larger class numbers, the clustering-based split still yields significantly higher $\text{MMD}(T, V)$ than both baselines.

## 6 Limitations

As with all robust optimisation, a "no free lunch" theorem (Wolpert & Macready, 1997) applies to model selection as well: using a worst-case split loses the guarantee that the selected model will be optimal in the nominal (ID) case. If no domain shift is expected, an ID validation set is clearly preferable; in general, robust strategies may be inappropriate if the uncertainty is minimal, so the nature of the problem should be considered prior to choosing a methodology. Additionally, if unlabelled test domain data is available at split time, this should also be factored in when choosing the validation set (e.g., using the method of Napoli & White (2025)). However, this is not applicable if the hyperparameters need to be decided before the test domain is accessible, e.g., for test-time adaptation, and our experiments show that the worst-case split is still valid in the UDA case. The motivations and considerations for using worst-case optimisation are discussed thoroughly in Sagawa et al. (2019). It is also noted that in the nominal case, the favourable training distribution will be the dominating factor on test accuracy compared to the model selection, and so sub-optimality of the selection method will be less of an issue.

On evaluation, using $\gamma = 1$ for the RBF kernel bandwidth was not an appropriate choice, especially in the high-dimensional case. A more practical option would be to use $\gamma = 1/d_F$, where $d_F$ is the dimensionality of the clustering features (Liu et al., 2020).

Dataset and training algorithm choices for these experiments were biased towards combinations with larger performance gaps between the random split and oracle criteria. This was necessary to properly validate the method: it would be impossible to see any significant signs of improvement if the random split and oracle criterion were within margin of error. Although this does inevitably limit the scope of the experiments, it

by no means invalidates the results: cluster-based splits *do* outperform random and metadata-based splits, *where such an improvement is possible.*

As stated in Section 1, a major advantage of the clustering-based validation split is that it is not dependent on domain metadata. However, even with the domain constraint removed or with metadata-independent training algorithms (e.g., empirical risk minimisation), the specific experimental setup in this paper precludes the ability to test this method in truly metadata-free settings. Due to the fundamental design of the DomainBed framework, domain metadata can still be leveraged (both implicitly and explicitly) through several avenues. For example, models are still trained on domain-balanced minibatches of data, and validation accuracy is averaged over the validation domains. The datasets themselves may also be unrealistically domain-balanced to begin with. However, the authors believe that these influences are mild enough that the overall trends observed in these experiments can reasonably be expected to hold in metadata-free settings as well.

The experiments would ideally be repeated across a range of model architectures to reflect the differences in generalisation power of larger/newer models. However, it was necessary to restrict the experiments in this paper to a single, smaller model (ResNet-18) due to the high computational costs involved in developing and comparing model selection criteria.

## 7 Conclusion

This paper presented a method for model selection under domain shift, where the training-validation split is performed using a constrained kernel k-means clustering algorithm. In addition to outperforming traditional methods, this approach is grounded by an observed strong correlation between the MMD between the training and validation sets, and test domain accuracy.

The algorithm is not parameter-free; future work could include a data-driven method for selecting these parameters, for example with an additional layer of meta-tuning. The algorithm can also trivially be extended to k-fold cross-validation by increasing the number of clusters. Finally, an investigation into the use of prior knowledge-informed feature extraction (see Appendix A) would be of interest.

## 8 Acknowledgements

This work was supported by grants from BAE Systems and the Engineering and Physical Sciences Research Council. The authors acknowledge the use of the IRIDIS High Performance Computing Facility, and associated support services at the University of Southampton, in the completion of this work.

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

# A Prior Knowledge-Informed Feature Extraction

Clustering-based splits may be rendered more effective if the features used for clustering are informed by domain knowledge on the underlying causes of the domain shift. In this section, we discuss various applications where domain knowledge may be available and propose relevant features which could leverage this information.

In visual tasks involving stylistic shifts, style features (Matsuura & Harada, 2020) or CLIP-derived tokens can explicitly capture these differences. In histopathology, tumour detection is affected by staining variations across tissue slides, so colour-based features such as hue histograms or colour deconvolution can be used. In wildlife monitoring, shifts due to habitat or camera placement can be isolated by using segmentation to extract backgrounds. Visual changes such as lighting or colour tone shifts can be captured through metrics like brightness, contrast, and colour temperature. In cases where image resolution or fidelity varies, edge density, compression or downsampling artefacts, or image quality scores may provide useful indicators. Audio tasks can suffer from variations in recording equipment or conditions; features such as spectral descriptors, microphone gain, and SNR estimates can help characterise these differences. Or, for speech recognition, speaker differences can be captured using accent classifiers, phonetic parameters, or word rate.

# B Derivation of equation (11)

Based on the law of total variance, a fundamental theorem in ANOVA is that the total sum of squares of a set of points can be decomposed into the sum of squares within each cluster and the sum of squares of each cluster centroid to the overall centroid (weighted by cluster size):

$$\text{SSq}(S_X) = \text{SSq}(T_X) + \text{SSq}(V_X) + |T_X| \, \|\mu_{\mathbb{P}_T} - \mu_{\mathbb{P}_S}\|_{\mathcal{H}}^2 + |V_X| \, \|\mu_{\mathbb{P}_V} - \mu_{\mathbb{P}_S}\|_{\mathcal{H}}^2 \, .$$

Next, using the fact that

$$\mu_{\mathbb{P}_S} = \frac{|T_X|\mu_{\mathbb{P}_T} + |V_X|\mu_{\mathbb{P}_V}}{|T_X| + |V_X|},$$

we have

$$
\begin{aligned}
\|\mu_{\mathbb{P}_T} - \mu_{\mathbb{P}_S}\|_{\mathcal{H}}^2 &= \left\| \mu_{\mathbb{P}_T} - \frac{|T_X|\mu_{\mathbb{P}_T} + |V_X|\mu_{\mathbb{P}_V}}{|T_X| + |V_X|} \right\|_{\mathcal{H}}^2 \\
&= \left\| \frac{\mu_{\mathbb{P}_T}(|T_X| + |V_X|) - |T_X|\mu_{\mathbb{P}_T} - |V_X|\mu_{\mathbb{P}_V}}{|T_X| + |V_X|} \right\|_{\mathcal{H}}^2 \\
&= \left\| \frac{|V_X|(\mu_{\mathbb{P}_T} - \mu_{\mathbb{P}_V})}{|T_X| + |V_X|} \right\|_{\mathcal{H}}^2 \\
&= \frac{|V_X|^2}{(|T_X| + |V_X|)^2} \, \|\mu_{\mathbb{P}_T} - \mu_{\mathbb{P}_V}\|_{\mathcal{H}}^2 \, .
\end{aligned}
$$

Similarly,

$$\|\mu_{\mathbb{P}_V} - \mu_{\mathbb{P}_S}\|_{\mathcal{H}}^2 = \frac{|T_X|^2}{(|T_X| + |V_X|)^2} \, \|\mu_{\mathbb{P}_T} - \mu_{\mathbb{P}_V}\|_{\mathcal{H}}^2 \, .$$

Substituting and simplifying:

$$
\begin{aligned}
\text{SSq}(S_X) &= \text{SSq}(T_X) + \text{SSq}(V_X) + \frac{|V_X||T_X|^2 + |T_X||V_X|^2}{(|T_X| + |V_X|)^2} \, \|\mu_{\mathbb{P}_T} - \mu_{\mathbb{P}_V}\|_{\mathcal{H}}^2 \\
&= \text{SSq}(T_X) + \text{SSq}(V_X) + \frac{|T_X||V_X|}{|T_X| + |V_X|} \, \|\mu_{\mathbb{P}_T} - \mu_{\mathbb{P}_V}\|_{\mathcal{H}}^2 \, .
\end{aligned}
$$

Finally, rearranging:

$$\|\mu_{\mathbb{P}_T} - \mu_{\mathbb{P}_V}\|_{\mathcal{H}}^2 = \frac{|T_X| + |V_X|}{|T_X||V_X|} (\text{SSq}(S_X) - \text{SSq}(T_X) - \text{SSq}(V_X)),$$

as required.

## C   Additional training and hyperparameter details

Table 6: General parameter values and training details for the experiments.

| Experimental parameter | Value |
|---|---|
| Hyperparameter random search size | 10 |
| Number of trials | 3 |
| Holdout fraction | 0.2 |
| UDA holdout fraction | 0.5 |
| Number of training steps | 3000 |
| Gaussian kernel bandwidth | 1 |
| Finetuning iterations before split | 3000 |
| Nyström subset size (if applicable) | 2000 |
| Architecture | ResNet-18 |
| Class balanced | True |

Table 7: Classification pipeline of the Humpbacks dataset. This follows the pipeline in (Napoli & White, 2023).

| Step number | Step detail |
|---|---|
| | **Acoustic front-end** |
| 1 | Resample to 10 kHz |
| 2 | Mel-scale filter bank with 64 filters |
| 3 | Short-time Fourier transform with 100 ms FFT window, 50% overlap |
| 4 | Per-channel energy normalisation |
| 5 | Split into 3.92 s (128 pixel) analysis frames with 50% overlap |
| | **CNN** |
| 1 | Conv2D (nodes=16, kernel=3x3, stride=2, activation=ReLU) |
| 2 | Conv2D (nodes=16, kernel=3x3, stride=2, activation=ReLU) |
| 3 | Conv2D (nodes=16, kernel=3x3, stride=2, activation=ReLU) |
| 4 | Conv2D (nodes=16, kernel=3x3, stride=2, activation=ReLU) |
| 5 | Global average-pooling 2D |
| 6 | Fully-connected layer |

## D   Worst-case accuracy validation

Table 8: Model selection using validation accuracy of the worst performing domain. Note that this does not apply to the leave-one-domain-out and oracle criteria, so these values are unchanged from Table 1.

| Split type | DG experiments | | | UDA experiments | | | Raw average | Normalised average |
|---|---|---|---|---|---|---|---|---|
| | Camelyon | Humpbacks | SVIRO | VLCS | PACS | TerraInc | | |
| Random | 83.2 ± 1.6 | 76.7 ± 3.1 | 98.2 ± 0.2 | 69.7 ± 3.0 | 81.8 ± 0.9 | 40.2 ± 1.9 | 75.0 ± 0.8 | 0.0 ± 13.5 |
| Leave-one-out | 85.6 ± 1.0 | 77.3 ± 1.7 | 98.6 ± 0.0 | 71.3 ± 3.5 | 83.7 ± 0.6 | 37.3 ± 2.5 | 75.6 ± 0.8 | 23.3 ± 12.1 |
| Linear k-means | 87.1 ± 0.2 | 77.8 ± 1.6 | 98.6 ± 0.1 | 73.7 ± 2.0 | 83.8 ± 0.6 | 42.3 ± 1.9 | 77.2 ± 0.5 | 49.8 ± 8.8 |
| RBF k-means | 87.5 ± 0.7 | 78.2 ± 0.6 | 98.5 ± 0.2 | 76.8 ± 0.6 | 83.9 ± 1.0 | 38.8 ± 2.5 | 77.3 ± 0.5 | 46.8 ± 10.9 |
| Mix-up | 83.9 ± 0.8 | 76.9 ± 1.5 | 98.4 ± 0.2 | 72.9 ± 1.5 | 81.0 ± 0.5 | 38.3 ± 2.7 | 75.2 ± 0.6 | 2.3 ± 10.7 |
| Oracle | 88.3 ± 0.6 | 85.4 ± 1.5 | 99.1 ± 0.1 | 77.6 ± 0.5 | 84.4 ± 0.4 | 45.8 ± 1.0 | 80.1 ± 0.3 | 100.0 ± 5.7 |

