# OpenReview forum: "Clustering-Based Validation Splits for Model Selection under Domain Shift"
_TMLR — Accepted by TMLR_

### Review · Reviewer_VAvi · 2025-05-07

**Summary Of Contributions:**

The authors has developed a method to make train/validation splits that --when used for model selection -- yields better performance out-of-distribution compared to baselines. They start idea that a model that performs well out-of-distribution should be one that performs minimax optimal over some set of evaluation distributions. They argue that one should find a validation set that is as dissimilar to the training set as possible in order to make the the selected model as robust as possible. They quantify the dissimilarity between train and validation, denoted MMD(T,V), develop a linear program that maximizes MMD(T,V), provide evidence about its effectiveness for model selection, and corroborates the heuristic that large MMD(T,V) improves Test performance.

**Audience:**

Yes

**Broader Impact Concerns:**

I do not think this work requires a broader impact statement. The compute use is reasonably low. The work seem to provide less fragile machine learning models, which I can only consider a positive impact.

**Claims And Evidence:**

Yes

**Requested Changes:**

1. Explain equation 1 better. Discuss it in relation to choice of kernel. Please also discuss choice of kernels in other parts of the article. In other settings, such as Covariate Shift by Kernel Mean Matching (Gretton et al 2008), the resuts are highly sensitive the kernel choice. Are your results also sensitive to kernel choice?
2. Discuss UDA/DG in introduction. Define the terms properly. How are they similar and different? I find this important.
3. Add discussion on class balance.
4. Fix the discussion on neural nets (see above). Not critical, but helps readability.
5. In section 1.4 it becomes clear that the target domain $\mathcal{Y}$ must be a finite set, i.e. classification problems. State this in the notation section 1.2 instead. Same for the domain $\mathcal D^S$ and $\mathcal D^E$ .
6. Before equation (5), you say ““This is equivalent to performing kernel k-means clustering”.  The reader would be helped much be a clear statement that k=2 (for T and V) at this place in the text. This is not critical but a good service to the reader.
6. Discuss runtime in relation to proposition 2. This is not critical, but would make the paper more appealing to others who might consider using the method.
7. Remove Figure 2 and the $\rho$ in the abstract. I believe that would improve the scientific standard of the article.

Two typographic points:
- Remove the abbreviations DG, DRO and UDA from the abstract. They are only defined and never reused.
- In equation (3), use $\mathcal{D}^S$, not $\mathcal D$. This seems to be a typo.

**Strengths And Weaknesses:**

The paper is well written. Clear language. Clear claims. Good illustrations. The idea is simple to follow. The authors address several concerns, such as empirical efficacy and computational needs. The method is general and widely applicable. It is interesting from both a ML perspective, and a statistical one. I appreciate that you report the GPU runtime for training. Well done!

The concerns I have with the paper are mostly about motivation and discussion. These points are listed below.

1. Equation 1 is provided and suggest that if MMD(T,V) is small, the validation loss will have a low upper bound, and provide little insight in robustness. It is not clear to the reader if this is valid under only some domain generalization problems (e.g. covariate shift) or much more generally. The reference Redko 2020 is new to me, and from a cursory read it was not obvious if it is only valid in covariate shift, or if it is valid more broadly. Equation 1 also depends on the kernel choice. A simple way to increase the MMD without altering the partition (T,V) is to use a different kernel, but obviously we will not get better predictors by changing the bandwidth in the RBF kernel and keeping the partition (T,V). That the influence of kernel choice is not discussed here or otherplaces in the paper is a shortcoming.
2. In section 1.4, the motivation for balanced classes is too brief. Enforcing balanced classes tends to give overly optimistic results. Why should we have class balance when optimizing for robustness with respect to domain shift?
3. On the middle of the page 4, the sentence “Moreover, the highest-level feature activations (those closest to the end of the network) are those most highly correlated with the network’s outputs […]” is the first and only mention of networks. Does the method only work with neural nets? If so, this must be said clearly. If not, just remove this confusing remark.
4. In proposition 2 you state the algorithm converges in finite number of steps. What is the time complexity? Not providing this makes the result feel weak.
5. In section 2 you start to contrast UDA and DG, and say that you design different experiments for the two settings. However, you never properly discussed and contrasted them in the introduction,  so it is hard for the reader to fully follow this part.
6. It is surprising that the related works section promotes Søgaard 2020 as a similar method, but you don’t include that in the benchmarking experiment. Why?
7. I think that the scientifically weakest part of the paper is Figure 2 and the accompanying significance test for rank correlation on pooled data. Since the patterns in Figure 1 are reasonably clear, I don’t see the point of Figure 2. The pooling after normalizing MMD seems statistically improper, so the computed $\rho = 0.63$ feels very shaky. The impression I get from the writing as it is now, is that you pooled the data to get higher statistical power and smaller p-value. That seems like the wrong thing to do.
8. Finally, you mention in the conclusion that prior knowledge is important. I think there should be a benchmark where prior knowledge is used in the train-val split. You would strengthen the paper by expanding this discussion, or even include that as a benchmark.

---

> ### Author Response · Authors · 2025-06-19
>
> Thank you for your feedback.
>
> > Questions about equation (1) (now equation (3) in the revised version) and kernel choice
>
> Equation (1) is not strictly limited to covariate shift. However, the robustness of the bound relies on the ability of the model to simultaneously perform well on both T and V, which can be affected by the presence of conditional shifts (and is captured in the $λ$ term).
>
> There are also assumptions made about both the (joint) data distribution and the hypothesis class of models – we refer to (Redko et al., 2022) for formal statements of these – and these ultimately define the reproducing kernel Hilbert space in which the MMD in the bound is measured (e.g., tightening/loosening the bound by changing the kernel bandwidth as you suggest also changes the assumptions made on the model and data). Thus, by symmetry, the kernel choice ultimately determines whether the MMD that we end up optimising is predictive of the mismatch between the training and validation losses.
>
> There are certainly applications in which the problem space is highly restricted (e.g., physical modelling) and implies a specific kernel choice. However, in our case, where the features are intermediate representations from a CNN, only relatively weak geometric assumptions can be made, which the RBF kernel tends to satisfy. It is also characteristic, so can distinguish between any two distributions. Perhaps more relevantly, we also note that the correlation between the MMD (with an RBF kernel) and R(θ, V) has been supported by empirical observations (Napoli & White, 2025).
>
> In this work, we fixed the RBF bandwidth to $γ = 1$ as a practical default, and verified by inspection that the induced kernel matrices had a good spread of values. To assess the sensitivity of our results to the choice of kernel, we also tested a linear kernel in both the main experiment (Table 1) and the ablation study (Table 2), and note that differences between the two are minor.
>
> Regarding your reference to kernel mean matching (KMM) (Gretton et al., 2008), we note that KMM uses the kernel distances directly to compute the importance weights, whereas kernel k-means clustering only compares between kernel distances when assigning the datapoints, and we would argue this makes the solution less sensitive to the kernel hyperparameters (and kernel choice overall).
>
> We have added this discussion to the revision.
>
> > Discuss UDA/DG in introduction.
>
> Detailed discussion and comparison added.
>
> > In section 1.4, the motivation for balanced classes is too brief. Enforcing balanced classes tends to give overly optimistic results. Why should we have class balance when optimizing for robustness with respect to domain shift?
>
> We think by “class balance” in this case you intend the clustering constraints on the size of each label group. These are present to ensure there are sufficient examples from each class in both T and V to properly train and validate the models. Without the constraints, the data would have a tendency to cluster by class, which may result in certain classes being placed entirely in one set. We have clarified this motivation in the paper.
>
> > Neural networks
>
> The line you quote is a consideration specifically for networks, which is the scenario we test the method with. But the method certainly is not limited to neural nets. We have clarified this in the paper.
>
> > Runtime
>
> Discussion of the runtime and some extra references to theory have been added.
>
> > It is surprising that the related works section promotes Søgaard 2020 as a similar method, but you don’t include that in the benchmarking experiment. Why?
>
> We only became aware of this reference relatively recently, which is why it was not tested in the main experiments.
>
> > Concerns regarding Figure 2
>
> We appreciate your concerns on this part of the paper, and have now removed this figure following your recommendation.
>
> > Finally, you mention in the conclusion that prior knowledge is important.
>
> We have added a detailed discussion of this in Appendix D. We are certainly interested in pursuing this avenue in future work.
>
> All remaining suggested changes have also been made.

---

> > ### Comment · Reviewer_VAvi · 2025-06-25
> >
> > Dear Authors,
> >
> > Thank you for making considerable changes and improvements. You did understand me correctly regarding "class balance" and I appreciate the expanded discussion about it.
> >
> > After further considerations, and reading both paper and code from Søgaard 2020, I understand their method would not be straight forward to benchmark against. The wasserstien distance is natural in their case, where the feature space is document term counts, but in your case, where you have features in $R^n$, one would need to choose a different metric. Their method further relies on cross validation, making it deviate from you method further.
> > Expanding slightly on this (one more sentence?) in the related works section could improve your article for readers like me, who are initially confused why you are not comparing with a method that you highlight as very similar.
> > This is not a critical change to make.

---

> > > ### Author Response · Authors · 2025-07-03
> > >
> > > Thank you for your comments. You are correct that Søgaard 2020 uses Wasserstein distance between distributions of term counts to characterise similarity between *individual* text documents, as opposed to *distributions* of documents. Thus, their method is somewhat less related than we had realised, and not at all straightforward to benchmark against. We have now clarified this in the latest revision, and thank you again for pointing this out.

---

### Review · Reviewer_oGTt · 2025-05-20

**Summary Of Contributions:**

The paper addresses model selection under domain shift, covering both unsupervised domain adaptation (UDA) and domain generalization (DG) scenarios. Motivated by principles from distributionally robust optimization (DRO) and domain adaptation theory, the authors propose a novel method for splitting the training data into training (T) and validation (V) sets. The key idea is to maximize the distributional discrepancy between T and V using Maximum Mean Discrepancy (MMD) as a divergence measure. They show that this objective is equivalent to performing kernel k-means clustering with k=2, where the two cluster centroids correspond to the T and V sets. To implement this, the paper introduces a constrained clustering algorithm that uses linear programming to control the size of the splits. It empirically shows that the proposed method outperforms existing splitting strategies across various datasets.

**Audience:**

Yes

**Broader Impact Concerns:**

I do not have any broader impact concerns for this paper.

**Claims And Evidence:**

Yes

**Requested Changes:**

1. Sections are not divided properly. I suggest to have Section 2. Prior work, Section 3. Preliminary (Theoretical motivation as a subsection), Section 4. Methodology
2. It would be better if there are equation numbers.
3. Suggest to use the capital $Y$ instead of $y$ for a restricted set e.g., $S_{X, y=y’} \rightarrow S_{X, Y=y’} $.
4. I expect to have answers for weakness 1, 2, 5-7, in the revision.

**Strengths And Weaknesses:**

Strengths
1. The writing of the paper is fairly clear and easy to follow.
2. The proposed method is simple but theoretically well motivated for domain adaptation tasks.
3. The paper provide detailed ablation studies for various scenarios.

Weaknesses
1. It assumes there are one cluster center for each T and V. How realistic assumption is that? Can the authors provide any argument for this?
2. Can Algorithm1 be extended to k > 2 case, which seems more realistic given that each of T and V consists of multiple classes, and may consist of multiple domains? If not, why?
3. I haven’t done any math but at the first glance, it seems like the objective of k > 2 is upper-bounded by $\Psi(T, V)$ (with modification for Eq.(3) accordingly). Would that justify a variant Algorithm1 with k > 2?
4. Every experiment result between RBF kernel k-means and Leave-one-domain-out is within one standard deviation
5. Limited experiments for larger-scale benchmark datasets such as DomainNet. I wonder  how good the proposed method is over baselines on datasets with a large number of classes like DomainNet given that only one cluster centroid is assumed for each of T and V.
6. How meaningful is MMD(T, V) in Figure1? According to the figure, MMD $< 5 \times 10^{-4}$, which seems fairly small to say there is a meaningful distinction between T and V from one algorithm over another.
7. In addition, can the authors also provide MMD(T, E) and MMD(V, E) along with MMD(T, V) to check if large MMD(T, V) potentially helps better generalizing on E?
8. I am not sure how meaningful the p-values are for Spearman's rank correlation given there are only 5 samples. If it is, can the authors provide a justification?
9. The authors said "The algorithm .... and comes with convergence guarantees." But I do not see convergence guarantees in the paper.

---

> ### Author Response · Authors · 2025-06-19
>
> Thank you for your feedback.
>
> (1-3)\. Using one centre per subset is derived from the definition of the MMD, which measures distance between the centres of T and V embedded in kernel feature space. The kernel k-means objective with k > 2 would be measuring pairwise MMDs between multiple subsets, and it is not clear to us how these would be combined back into the 2 subsets required, or whether this would provide any advantage in the end. k > 2 would be used if performing k-fold cross validation, in which case k would be set accordingly (this is already discussed in the paper). The fact that the data may have a tendency to cluster by class is unwanted in our application and we address this with the per-class clustering constraints.
>
> 4\. We have mentioned this in the paper
>
> 5\. We agree that more classes may have a detrimental effect on the method (but perhaps not for the reason you suggest). In general, with more classes, the objective function can be reduced less due to the increased number of clustering constraints. For example, in an extreme case with only 2 examples per class, a random split (stratified by class) and a clustering-based split would be equivalent.
> Unfortunately, for both of the larger Domainbed datasets (Office-Home and DomainNet), performance of the random split and oracle criterion are within margin of error (this can be seen from the results in (Gulrajani & Lopez-Paz, 2021)) so it is not possible to use these to validate our method (this limitation has already been mentioned in the paper). Nonetheless, as this is not something we had previously considered, we have added a discussion of this effect in the Limitations section, and we hope this is enough to address your point.
>
> 6\. We can point to two effects which suggest that the differences between the MMD values are meaningful. Firstly, the standard errors across the experimental repeats are smaller than the differences between datapoints (they are all at least 2-3 orders of magnitude smaller than the data range – we have now added a discussion of the standard errors to the manuscript). Additionally, if the differences were below the noise floor, then no correlation would be detectable. Instead, the correlation is evident.
> The reason the MMD scales are so small for some of the subplots is an effect of our choice to keep the kernel bandwidth fixed for all datasets.
>
> 7\. If we understand the motivation of your question correctly, you want to check that the worst-case MMD(T,V) achieved by our method is greater than MMD(T,E) or MMD(V,E) and thus that our DRO formulation is valid.
> Unfortunately, we did not store these values when running our main experiments, but we have been able to conduct some new small-scale simulations on a single dataset (a Humpbacks subset of 8,000 examples) which we hope is sufficient to address your question (but note these are not comparable with the results in the paper). The below table shows the MMDs averaged across 10 repeats and all 4 domains of the dataset. We can see that the RBF kernel k-means split is the only one for which MMD(T,V) exceeds MMD(T,E) and MMD(V,E).
>
>
> |     Method                |     MMD(T,V)         |     MMD(T,E)       |     MMD(V,E)       |
> |---------------------------|----------------------|--------------------|--------------------|
> |     Linear k-means        |     0.121±0.015      |     0.120±0.078    |     0.130±0.074    |
> |     RBF kernel k-means    |     0.178±0.03       |     0.146±0.079    |     0.132±0.089    |
> |     Random                |     0.0006±0.0001    |     0.095±0.064    |     0.095±0.064    |
> |     Leave 1 domain out    |     0.0518±0.034     |     0.053±0.036    |     0.135±0.087    |
>
> 8\. You are correct that the p-values, if only considered individually, are quite large and make it difficult to claim that the correlation is significant. However, considering all 6 subplots together we see that the correlation is consistent across 5 of the 6 cases, which raises our confidence. This was our motivation for combining the data in Figure 2, although we have now removed this due to concerns from another reviewer.
>
> 9\. Discussion of the convergence is in Proposition 2, page 6.
>
> Suggested notation changes: these have now been made.

---

> > ### Comment · Reviewer_oGTt · 2025-06-25
> >
> > Thank you for the clarification. The rebuttal has addressed many of my initial concerns. However, I still have a concern for the effectiveness of the proposed method.  As noted in point 5, the authors acknowledge that the method would yield smaller improvements on datasets with a larger number of classes. Furthermore, as mentioned in point 4, even on datasets with fewer classes (as used in the experiments), the observed gains are modest; often within one standard deviation. Although the authors have clearly specified these limitations, I am still unsure about the practical utility of the method, particularly given its non-negligible computational overhead and the difficulty in anticipating when it will be effective for an unseen dataset. In particular, it remains unclear what the threshold is for the number of classes beyond which the method becomes ineffective. Could the authors provide further justification or guidance on when the method is expected to be beneficial in practice e.g. the number of classes of datasets?

---

> > > ### Author Response · Authors · 2025-07-01
> > >
> > > Thank you again for your valuable feedback. To address your remaining concern, we have designed another experiment to analyse the effect of the number of classes on MMD(T,V). Specifically, we use the same Humpbacks features as our response to point 7 above, and generate synthetic class labels using another round of kernel k-means clustering. Then, we perform our data split using the synthetic labels. The below table shows how the MMDs change as the number of classes varies between 1 and 500. Note that MMD(T,E) and MMD(V,E) also tend to reduce as the number of classes increases due to the triangle inequality.
> > >
> > > |     Number of classes    |        MMD(T,V)      |        MMD(T,E)      |        MMD(V,E)      |
> > > |:------------------------:|:--------------------:|:--------------------:|:--------------------:|
> > > |             1            |     0.221 ± 0.016    |     0.152 ± 0.042    |     0.148 ± 0.040    |
> > > |             2            |     0.187 ± 0.007    |     0.154 ± 0.048    |     0.129 ± 0.048    |
> > > |             5            |     0.177 ± 0.007    |     0.121 ± 0.037    |     0.157 ± 0.034    |
> > > |             10           |     0.161 ± 0.006    |     0.132 ± 0.020    |     0.138 ± 0.021    |
> > > |             20           |     0.159 ± 0.005    |     0.132 ± 0.021    |     0.137 ± 0.019    |
> > > |             50           |     0.145 ± 0.002    |     0.137 ± 0.031    |     0.125 ± 0.031    |
> > > |            100           |     0.138 ± 0.004    |     0.119 ± 0.019    |     0.140 ± 0.017    |
> > > |            500           |     0.129 ± 0.001    |     0.123 ± 0.027    |     0.130 ± 0.027    |
> > >
> > > To determine the maximum number of classes for which our method is still valid, we can consider the range for which MMD(T,V) is higher than MMD(T,E) and MMD(V,E), since this is the assumption made by DRO. We consider both MMD(T,E) and MMD(V,E) since the data split is symmetric (it does not matter which way around T and V are labelled). In this case, we can see from the table that the threshold occurs at around 100 classes.
> > >
> > > Although this value may be dataset-dependent, we do not think it is unreasonable that if a user is close to this threshold or wants to be sure they are getting the largest possible MMD, they simply try multiple splitting methods on their dataset and pick the split for which MMD(T,V) is highest. We also note that even for larger class numbers, our clustering-based split still yields significantly higher MMD(T,V) than the metadata-based split – see the table in our previous response.
> > >
> > > We will add full details of this experiment to the manuscript in the coming days, and hope that this answers your query.

---

### Review · Reviewer_UUVW · 2025-06-06

**Summary Of Contributions:**

The authors consider the problem of model selection under domain shift. More precisely, they assume that the data at hand comes from several domains, and that the model will be evaluated on yet another domain. The authors argue that standard random training/validation splits will lead to selecting hyperparameters that are not robust to the shift, and suggest using a split based on constrained kernel $k$-means.

The main motivation for using kernel $k$-means is that this will maximise the maximum mean discrepancy (MMD) between the training and validation sets. The constraints on the kernel $k$-means are here to make sure that the training and validation sets have the same labels/domains proportions.

Their experiments on several datasets indicate that the $k$-means splits allow for selecting better hyperparameters than standard random splits, as well as other splitting strategies.

**Audience:**

Yes

**Broader Impact Concerns:**

I do not have particular concerns here.

**Claims And Evidence:**

Yes

**Requested Changes:**

I believe it is very important to act upon the points 1. and 2. above. Adding the baseline of point 3. is less pressing, but would improve the paper.

**Strengths And Weaknesses:**

# Strengths

- While a significant body of work has been done on robust inference under domain shift, hyperparameter selection has beens seemingly quite overlooked, and I think that this paper somehow helps filling this gap (I am not an expert of the domain so I may be missing some references).

- The general story of maximising the MMD to be more robust, and using $k$-means to that end is quite convincing, and well-narrated.

- The experiments are very convincing, I particularly enjoyed the attention to detail in the ablation study and in the attempts at assessing the effects of training versus validation (Section 2.5). I do believe that a simple baseline (stratified splitting) could make them even more convincing though (see weaknesses).

# Weaknesses

My main concerns are related to the discussion of prior work:
1. Most importantly, the authors mention that their technique is very similar to the one of Wecker et al. (2020), but the differences and similarities between the two approaches are not clear enough. A detailed discussion (potentially with small experiments, and potentially in the Appendix) of the different motivations/algorithms/contributions of the two papers would be very beneficial.
2. Less importantly, the main theoretical contribution (the link between MMD and $k$-means) is not really new, and related results were obtained (without the same constraints) by França et al. (2020) and Ohl et al. (2024). Here again, discussing these results would improve the paper.

As mentioned earlier, I really enjoyed the experiments, but have the small following comment about them:

3. A very natural baseline would be to use stratified splits (to satisfy the same constraints as $k$-means). This can be done using, e.g. the model selection module of Scikit-learn.

# Minor points

- In the MMD experiments, why are there only $5$ datapoints? I thought experiments were repeated?

# Very minor points

- Some (admittedly very well-known) acronyms are not defined: AI, ERM
- The authors use Theorem 36 from Redko et al. (2020). The paper was updated on arXiv in 2022, but fortunately, Theorem 36 is still Theorem 36. It would be nice to cite the last version (and more importantly to check if it is updated again).

# References mentioned in the paper

- Redko et al., A survey on domain adaptation theory: learning bounds and theoretical guarantees, arXiv preprint 2022
- Wecker et al., ClusterDataSplit: Exploring Challenging Clustering-Based Data Splits for Model Performance Evaluation,  Proceedings of the First Workshop on Evaluation and Comparison of NLP Systems, 2020

# Additional references

- França et al., Kernel k-Groups via Hartigan’s Method, IEEE TPAMI, 2020
- Ohl et al., Kernel KMeans clustering splits for end-to-end unsupervised decision trees, arXiv preprint 2024

---

> ### Author Response · Authors · 2025-06-19
>
> Thank you for your feedback.
>
> 1. We have expanded the discussion of the relation of our paper with Wecker et al. (2020), and also conducted some small simulations to show our method has improved convergence (Section 5.6).
>
> 2. Thank you for providing these references, they have been added to the text.
>
> 3. Apologies for the confusion. Our “random split” baseline is already stratified by domain and the accuracy is class-balanced, so this is (in expectation) equivalent to the “stratified splits” method you mention. We have made it clearer in the experimental description that this is the case.
>
> > In the MMD experiments, why are there only 5 datapoints? I thought experiments were repeated?
>
> Each datapoint represents the mean across the repeats and test domains. We have clarified this in the paper.
>
> Very minor points: both fixed.

---

> > ### Comment · Reviewer_UUVW · 2025-07-01
> >
> > Thanks, my concerns have been addressed accordingly!

---

### Decision · Action_Editor_tzZp · 2025-07-31

**Recommendation:** Accept with minor revision

**Additional Comments:**

Although there is no question regarding the kernel choice, AE wants to mention that choosing gamma as 1 is not practical, especially in the high-dimensional case. Normally, you can choose gamma as 1/d, where d is the dimension of the date fed into the kernel function (you can refer to the deep kernel for the two-sample testing paper).

**Audience:**

Yes

**Audience Explanation:**

The studied problem is quite important for the domain adaptation field: model selection. Because we normally do not have labeled data in DA settings, it is important to know how to select models. This paper made good progress toward this research direction.

**Claims And Evidence:**

Yes

**Claims Explanation:**

This paper is carefully written to ensure all claims are well supported by the evidence. All relevant literature is included, and reviewers/readers can easily judge the main contribution of this paper. There is a minor comment from AE demonstrated in the additional comment, which can further broaden the practicality of this paper.

---

> ### Author Response · Authors · 2025-08-11
>
> Thank you for your additional comment, we have added this to the Limitations section